# Sex differences in the association between area deprivation and generalised anxiety disorder: British population study

Olivia Remes, Nick Wainwright, Paul Surtees, Louise Lafortune, Kay-Tee Khaw, Carol Brayne

▶ Prepublication history and additional material is available. To view please visit the journal (http://dx.doi.org/10.1136/bmjopen-2016-013590).

Department of Public Health and Primary Care, University of Cambridge, Cambridge, UK

**Correspondence to**
Olivia Remes;
or260@medschl.cam.ac.uk

## ABSTRACT

**Objective:** Studies have shown that area-level deprivation measured by factors, such as non-home ownership, non-car ownership and household overcrowding, can increase the risk for mental disorders over and above individual-level circumstances, such as education and social class. Whether area-level deprivation is associated with generalised anxiety disorder (GAD) independent of personal circumstances, and whether this association is different between British women and men is unknown.

**Design:** Large, population study.

**Setting:** UK population-based cohort.

**Participants:** 30 445 people from the general population aged 40 years and older and living in England consented to participate at study baseline, and of these, 21 921 participants completed a structured health and lifestyle questionnaire used to capture GAD. Area deprivation was measured in 1991 using Census data, and GAD was assessed in 1996–2000. 10 275 women and 8219 men had complete data on all covariates.

**Main outcome measure:** Past-year GAD defined according to the Diagnostic and Statistical Manual of Mental Disorders, fourth edition (DSM-IV).

**Results:** In this study, 2.5% (261/10 275) of women and 1.8% (145/8219) of men had GAD. Women living in the most deprived areas were over 60% more likely to develop anxiety than those living in areas that were not deprived (OR=1.63, 95% CI 1.21 to 2.21; p=0.001), but this association between deprivation and GAD was not apparent in men (OR=1.13, 95% CI 0.72 to 1.77; p=0.598).

**Conclusions:** The absolute numbers of people living in deprived conditions are large worldwide. This, combined with a growing mental health burden, means that the findings obtained in this study remain highly relevant. The WHO has emphasised the need to reduce social and health inequalities. Our findings provide a strong evidence base to this call, showing that the environment needs to be taken into account when developing mental health policy; gender is important when it comes to assessing the influence of the environment on our mental health.

### Strengths and limitations of this study

- We used a large, population-based sample of middle-aged and older-aged adults and adjusted for a range of important confounders, such as sociodemographic factors and medical history.
- We used a structured, self-reported questionnaire to assess the presence of past-year generalised anxiety disorder, and participants were followed for 7 years.
- We measured area deprivation by employing a commonly used and theoretically sound index.
- Those who participated in this study were somewhat less deprived and healthier than individuals living in other parts of England; therefore, our results may not generalise to people living in extremely deprived circumstances.

## INTRODUCTION

Generalised anxiety disorder (GAD)[1] is a common and persistent disorder, and is associated with increased risk for disability and suicide.[2–5] GAD can lead to serious impairment in social and occupational functioning, and once it develops, it increases the risk for major depression, substance misuse and serious physical medical conditions.[5–8] This disorder has a chronic course and is difficult to treat.[5] Consequently, it is important that its risk factors are identified for prevention and targeted intervention.

Few studies have assessed the risk factors of GAD; therefore, information is scarce. The studies that have been undertaken have focused on characteristics measured at the level of the individual, such as personal income and education,[9–11] demographics[12 13] and family history of psychopathology.[13] However, research has shown that the living context, such as area deprivation, can have profound effects on health, independent of personal characteristics.[14–16] Area deprivation

refers to residential environments or living contexts characterised by factors, such as high levels of unemployment, non-home ownership, non-car ownership and low income.[14]

Many studies conducted in western countries have shown that living in areas characterised by high-income inequality can lead to significantly increased risks for serious medical conditions and mortality.[15] [16] A meta-analysis of cohort studies showed that people living in areas of high-income inequality, as measured by the GINI index, had an increased risk for mortality.[17] Population-based studies further showed that living in disadvantaged neighbourhoods or places where there is high chronic stress can increase the risk for mental disorders, such as depression.[18–20] Whether area deprivation can be used to predict GAD is unknown.

In this population-based, cohort study, we examine the association between area deprivation and GAD, while controlling for a number of confounders, including previous medical conditions, major depressive disorder and sociodemographic factors. Results are presented separately for women and men, and this is performed for several reasons. Research has shown that women are more likely to develop anxiety compared with men, mainly due to genetic and hormonal factors, social roles or gender norms and environmental factors.[21–23] Gender has been linked to resources derived from the environment.[21] [22] Compared with men, women have been shown to have less access to material resources and social status positions, and this can influence mental health. Women also seem to interact with their environment differently. For example, women are exposed to different stressors compared with men, because of gender differences with respect to social roles.[18] [21]

Despite these differences, research examining the link between the living context, such as area deprivation, and mental health among women and men, separately is scarce. It remains unclear whether there are sex differences in the association between area deprivation and risk of GAD—and our objective in this study will be to assess this. Knowing that one sex is at risk of developing anxiety when exposed to deprived circumstances helps to tailor interventions and allocate scarce resources according to need.[24]

## METHODS

Data were drawn from the European Prospective Investigation of Cancer (EPIC)-Norfolk, whose design and study methods have been described in detail elsewhere.[25] In brief, a prospective population-based cohort of 30 445 participants aged 40–74 years were recruited by post between 1993 and 1997 through general practice age–sex registers in the city of Norwich and the surrounding small towns and rural areas (77 630 people were initially invited to join EPIC-Norfolk). At baseline (1993–1997), 30 445 participants consented to join the study and completed a postal Health and Lifestyle (HLQ) questionnaire that captured information on sociodemographics, including sex, marital status, highest educational attainment and self-reported physician diagnoses of physical diseases. Using participants' postal codes, a measure of area deprivation was derived based on the 1991 Census. Social class was also obtained from the Census. Between 1993 and 2000, participants completed self-reported postal questionnaires, provided they: (1) were still alive, (2) did not ask to be removed from the study's mailing list and (3) had a valid mailing address.

During 1996–2000, 20 921 participants completed a structured, psychosocial Health and Life Experiences (HLEQ) questionnaire. During this time, an assessment of GAD and major depressive disorder (MDD) was made according to the Diagnostic and Statistical Manual of Mental Disorders, fourth edition (DSM-IV)[1] [26] (figure 1). Using the HLEQ questionnaire, age and then disability measures based on the SF-36 were also derived.[27]

All participants recruited through general practice registers and who completed a baseline health questionnaire were eligible to be included in our study; those

**Figure 1** Flow chart of European Prospective Investigation of Cancer (EPIC)-Norfolk cohort. This is a flow chart showing the number of participants at each study stage: the number approached to participate in the EPIC-Norfolk study, the number enrolled at baseline, and with complete data on all covariates. The EPIC-Norfolk study consists of middle-aged and older British people.

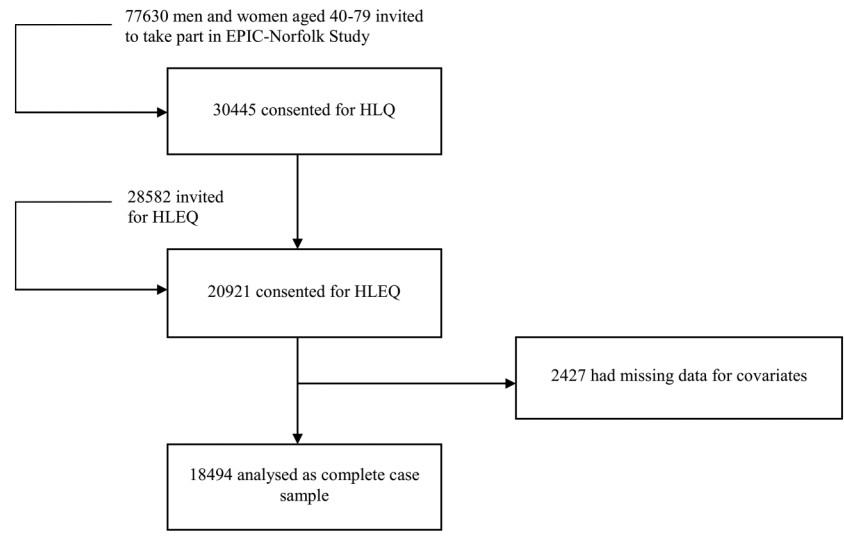

who completed a psychosocial questionnaire during follow-up were eligible to be included in our analysis.

## Dependent variable

The primary outcome in this study was past-year GAD. The self-reported HLEQ questionnaire captured the onset and offset timings of episodes of past-year GAD.[26] Past-year GAD consisted of at least one episode that had offset within 12 months of administration of the HLEQ. DSM-IV GAD was present if participants reported having uncontrollable, excessive worry for 6 months or longer on most days than not that resulted in disability or impairment. In addition, at least three of the following symptoms needed to have been present: restlessness, irritability, muscle tension, fatigue, trouble concentrating because of worry, mind going blank, trouble falling asleep, trouble staying asleep and feeling keyed up or on edge.

## Individual-level measures (potential confounders)

Individual-level measures included age, sex, social class, marital status and educational level. The final categorisation of the variables took cell size into account and was also performed in accordance with previous literature.[26] [28–33] Social class was derived using the Computer-Assisted Standard Occupational Coding[34] and categorised as follows: (1) professionals, (2) managerial and technical occupations, (3) non-manual and manual (skilled workers), (4) partly skilled workers and (5) unskilled manual workers. To assign social class to men and women, the male partner's current or past occupation was used. If this information was not available, the female partner's occupation was used. If the social class from either partner was unavailable, then it was coded as missing. The final categorisation of social class included manual: skilled manual, partly skilled and unskilled; and non-manual: professionals, managerial and technical, and skilled non-manual. Marital status was categorised into three groups: married, single (or never married) and others (widowed, divorced, separated). Educational attainment was categorised into high (vocational or formal qualifications at the A-level or O-level or degree-level qualifications) versus low (no formal qualifications).

Individual-level health status was assessed through the construction of a variable capturing major prevalent physical diseases associated with anxiety.[35] This was based on HLQ questions asking participants: 'Has the doctor ever told you that you have any of the following?', followed by a list of options, such as allergies, asthma, cancer, stroke, heart attack, diabetes, thyroid conditions, etc. To determine disability levels, we used the physical component summary score (PCS) of the Medical Outcomes Study 36-Item Short Form (SF-36), a widely used, validated self-assessment tool. Higher scores indicate better health. PCS scores were dichotomised above and below the median. Lifetime MDD was assessed using the HLEQ.[26]

All of these individual-level variables were regarded as potential confounders and selected based on the literature and their association with anxiety[35–38] and deprivation.[39 40]

## Area-level measure (exposure variable)

To examine area deprivation, we used one of the most commonly used measures of area deprivation in the UK: the Townsend index.[41 42] This index is a composite measure of four variables obtained from the 1991 Census: (1) percentage of economically active residents over age 16 who are unemployed, (2) percentage of households that do not possess a car, (3) percentage of private households that are not owner-occupied, and (4) percentage of private households that are overcrowded (have more than 1 person per room). These variables were obtained at the level of the enumeration district. These four factors were then standardised by deriving Z scores (dividing the mean by the SD across enumeration districts in England and Wales). The Z values of the four variables were added together to produce a Townsend index score for each enumeration district. Positive values of the index indicate enumeration districts that are more deprived, while negative values indicate those that are less deprived; 0 represents the national mean. The postal codes of participants were record linked to enumeration districts, and participants were considered to live in deprived areas depending on the Townsend index score assigned to their enumeration district.[41]

The Townsend deprivation index was also disaggregated into its four constituent components to determine whether any one of these four is associated with GAD or if it is the combined components that matter.

## Statistical analysis

Characteristics of the participants were compared by GAD status. We used correlated data analysis to assess the association between individual-level and area-level risk factors of GAD. A population-average model was constructed, which accounted for the potential correlation introduced by the clustering of individuals within enumeration districts. To estimate the population-average effect of the risk factors of interest on past-year GAD, we used generalised estimating equations. As past-year GAD represents a binary outcome (yes/no) and the intracluster correlation is assumed to be equal, GEE with a logit link and an exchangeable correlation structure was used. Adjusted ORs and 95% CIs based on robust SEs were estimated. Standard multivariate logistic regression was also conducted and compared with the findings based on GEE.

Individual-level measures consisted of sociodemographic and health-related variables, whereas the area-level measure comprised the Townsend index. Townsend index scores were used to create a dichotomous variable, with 0 as the cut-point (representing the national average). Similarly, when the Townsend index

was disaggregated into its four consistent components, each variable was dichotomised using 0 (the national average) as the cut-point.

Analyses were conducted separately for men and women. First, unadjusted effect estimates were determined. Next, models were constructed to adjust for (1) age, social class, educational attainment; then for (2) age, social class, educational attainment, lifetime history of MDD; and finally for (3) age, social class, educational attainment, lifetime history of MDD, physical diseases and disability level. Age was first assessed as a categorical variable, and subsequently divided into 10-year bands. Models were constructed for participants with complete measurements on all covariates. It was not possible to group the GAD variable otherwise since it was created and categorised according to the DSM-IV,[26 43] and area deprivation was analysed in accordance with the literature.[31 44] In a subsequent analysis, a fully adjusted model was built in which the Townsend index was replaced by its four constituent components to determine whether any one of these four variables is significantly associated with GAD.

Finally, analyses were run with GAD without MDD as the outcome, in which past-year MDD was excluded. All models used two-sided statistical tests and a p value of <0.05 was considered statistically significant. Analyses were implemented in Statistical Analysis Software (SAS) V.9.3 (SAS Institute, Cary, North Carolina, USA).

To arrive at the study size, we went through the following steps: of the 30 445 who completed the baseline HLQ, we retained those participants who completed the HLEQ (20 921), and of these, we kept those people with complete data on all covariates (18 494) (figure 1).

### Patient involvement

There were no patients involved in the development of the research question and outcome measures, the design of the study, or the recruitment to and conduct of the study.

### RESULTS

A total of 77 630 people from general practices in Norfolk were invited to take part in the study, and of these, 30 445 consented. The characteristics of responders versus non-responders are compared in online supplementary appendix I; compared with non-responders, those who took part consisted of slightly more women and slightly younger participants. Of the 30 445 people recruited at baseline, 20 921 completed the HLEQ during follow-up. Of those who completed the HLEQ, 18 494 (88.4%) were available for analysis in this study, because they had data on all covariates. The number of missing observations for each covariate was: 9 for education, 47 for marital status, 497 for GAD, 468 for MDD, 458 for social class, 75 for the Townsend index and 1386 for the SF-36. Participants were followed between 1993 and 2000 (7 years).

The study sample consisted of a total of 10 275 women and 8219 men over the age of 40. Table 1 shows the distribution of individual-level and area-level characteristics by past-year GAD.

The overall prevalence of past-year GAD was 2.5% (261/10 275) for women and 1.8% (145/8219) for men. Women and men with GAD were <50 years of age, of higher educational attainment, single, in non-manual occupations, with prevalent physical diseases, higher levels of disability and MDD (table 1).

Findings from the correlated data analysis showed that the risk of GAD in women living in the most deprived areas was over 70% higher than in those living in the least deprived areas, even after adjusting for age and sociodemographics (OR=1.77, 95% CI 1.33 to 2.36) (table 2).

The OR reduced slightly after additionally controlling for MDD (OR=1.65, 95% CI 1.23 to 2.22, p=0.001), but remained significant. A strong association was present after further adjusting for prevalent physical diseases and disability (OR=1.63, 95% CI 1.21 to 2.21; p=0.001). To further determine the aspect of deprivation that is specifically related to GAD in women, the four separate components of the Townsend index were included in a fully adjusted model. Results showed that the effect estimates were highest for non-car ownership (OR=1.46, 95% CI 0.98 to 2.17; p=0.061), followed by non-home ownership (OR=1.27, 95% CI 0.87 to 1.86; p=0.222) and were lowest for unemployment (OR=1.07, 95% CI 0.76 to 1.52; p=0.694) and overcrowding (OR=0.75, 95% CI 0.53 to 1.07; p=0.111); these variables did not reach statistical significance.

In men, no association existed between anxiety and area deprivation in unadjusted and adjusted analyses (model C OR=1.13, 95% CI 0.72 to 1.77; p=0.598) (table 3).

We had similar findings when logistic regression was used in these models instead of GEE, suggesting that the intraclass correlation is negligible (findings not shown).

To assess whether deprivation was associated with past-year GAD without MDD in women, we excluded participants reporting past-year MDD (while controlling for all covariates in a fully adjusted model). Deprivation continued to be strongly associated with past-year GAD (OR=1.61, 95% CI 1.06 to 2.43) (findings not shown). In men, the association was still statistically non-significant (OR=1.34, 95% CI 0.73 to 2.47).

### DISCUSSION

In this analysis of data from a population-based, cohort study, we show, for the first time, that area deprivation is significantly associated with increased risk for GAD in women, but not in men. The association in women was independent of characteristics measured at the level of the individual, including sociodemographics and major medical conditions. When we assessed the specific

**Table 1** Distribution of characteristics for women (n=10 275) and men (n=8219) who completed the Health and Life Experiences questionnaire in the European Prospective Investigation of Cancer-Norfolk cohort

| Characteristic | Women | | Men | |
|---|---|---|---|---|
| | Number with characteristic | Percentage and number with past-year GAD | Number with characteristic | Percentage and number with past-year GAD |
| *Individual-level variables* | | | | |
| Sociodemographics | | | | |
| Age (years) | | | | |
| <50 | 1444 | 3.7 (54) | 961 | 3.2 (31) |
| 50-60 | 3693 | 3.2 (119) | 2645 | 2.4 (63) |
| 60-70 | 3167 | 1.9 (61) | 2739 | 1.2 (33) |
| >70 | 1971 | 1.4 (27) | 1874 | 1.0 (18) |
| Education* | | | | |
| Low | 4030 | 2.1 (83) | 2363 | 1.7 (39) |
| High | 6245 | 2.9 (178) | 5856 | 1.8 (106) |
| Marital status | | | | |
| Single | 414 | 3.1 (13) | 302 | 4.0 (12) |
| Married | 7714 | 2.4 (183) | 7221 | 1.5 (111) |
| Other† | 2147 | 3.0 (65) | 696 | 3.2 (22) |
| Social class‡ | | | | |
| Manual | 3820 | 2.3 (89) | 3281 | 1.7 (55) |
| Non-manual | 6455 | 2.7 (172) | 4938 | 1.8 (90) |
| Health status | | | | |
| Prevalent physical disease§ | | | | |
| Yes | 5660 | 3.1 (174) | 3836 | 2.2 (86) |
| No | 4615 | 1.9 (87) | 4383 | 1.4 (59) |
| Disability level | | | | |
| High¶ | 5258 | 3.3 (172) | 4009 | 2.6 (104) |
| Low | 5017 | 1.8 (89) | 4210 | 0.97 (41) |
| Lifetime MDD | | | | |
| Yes | 1926 | 8.7 (167) | 934 | 10.0 (93) |
| No | 8349 | 1.1 (94) | 7285 | 0.7 (52) |
| *Area-level variable* | | | | |
| Townsend index | | | | |
| Deprivation | | | | |
| Yes (>0) | 1636 | 3.9 (64) | 1237 | 2.3 (28) |
| No (≤0) | 8639 | 2.3 (197) | 6982 | 1.7 (117) |

*High education: O-level, A-level, degree; low education: refers to no education.
†Other: divorced, separated, widowed.
‡Manual: skilled manual, semiskilled, non-skilled; non-manual: professionals, managerial, skilled non-manual.
§Prevalent physical disease: respiratory disease (asthma and bronchitis), allergies (allergies and hay fever), stroke, heart attack, cancer, diabetes, thyroid conditions, arthritis.
¶Below the median PCS value of 50.6.

aspects of deprivation associated with anxiety in women, we found that those living in areas characterised by a high level of non-car ownership and non-home ownership were at increased risk of GAD, although the associations were not statistically significant. It appears that it is the overall effect of living in deprivation rather than a particular aspect of the living context that is associated with a statistically significantly increased risk of anxiety in women. It is difficult to show causality between area deprivation and GAD; however, a rigorous analysis based on cohort data is a an acceptable method of examining this relationship. The analysis was rigorous, because we used reliable and commonly used measures of area deprivation and GAD, controlled for covariates that are associated with the exposure (area deprivation) and outcome (GAD), had access to a large sample size of

over 18 000 people and followed participants for a long period (7 years).

### Potential mechanisms

The context as measured by Census composite deprivation indices appears to have a different relationship with the mental health of women and men, even after adjusting for individual socioeconomic status, demographics and other psychiatric and major medical conditions. Several mechanisms can account for this. Women perceive, relate to and engage differently from men.[45 46] Women are more exposed to the living context perhaps due to their greater uptake of part-time work and domestic or childrearing duties.[47] Since they are more embedded in their neighbourhoods, they are also more likely to be exposed to the stress that comes with living

**Table 2** ORs for past-year generalised anxiety disorder according to individual-level and area-level characteristics for women (n=10 275) who completed the Health and Life Experiences questionnaire in the European Prospective Investigation of Cancer-Norfolk cohort

| Characteristic* | ORs and 95% CI | | | | p Value for Model C |
| --- | --- | --- | --- | --- | --- |
| | Unadjusted | Model A† | Model B‡ | Model C§ | |
| *Individual-level variables* | | | | | |
| Sociodemographics | | | | | |
| Age (per 10 years) | 0.65 (0.56 to 0.74) | 0.63 (0.54 to 0.73) | 0.73 (0.62 to 0.85) | 0.66 (0.56 to 0.77) | <0.0001 |
| Education¶ | | | | | |
| Low | 0.72 (0.55 to 0.93) | 0.85 (0.64 to 1.12) | 0.90 (0.68 to 1.20) | 0.90 (0.68 to 1.20) | 0.475 |
| High | 1.00 | 1.00 | 1.00 | 1.00 | |
| Marital status | | | | | |
| Single | 1.33 (0.75 to 2.36) | 1.31 (0.73 to 2.36) | 1.36 (0.74 to 2.50) | 1.34 (0.73 to 2.47) | 0.348 |
| Married | 1.00 | 1.00 | 1.00 | 1.00 | |
| Other** | 1.28 (0.96 to 1.71) | 1.48 (1.09 to 2.00) | 1.09 (0.80 to 1.48) | 1.07 (0.79 to 1.46) | 0.671 |
| Social class†† | | | | | |
| Manual | 0.87 (0.67 to 1.13) | 0.89 (0.68 to 1.17) | 0.89 (0.68 to 1.18) | 0.85 (0.64 to 1.13) | 0.271 |
| Non-manual | 1.00 | 1.00 | 1.00 | 1.00 | |
| *Health status* | | | | | |
| Lifetime MDD | | | | | |
| Yes | 8.34 (6.44 to 10.79) | | 7.55 (5.78 to 9.86) | 7.00 (5.34 to 9.17) | <0.0001 |
| No | 1.00 | | 1.00 | 1.00 | |
| Prevalent physical disease‡‡ | | | | | |
| Yes | 1.65 (1.27 to 2.14) | | | 1.43 (1.09 to 1.88) | 0.011 |
| No | 1.00 | | | 1.00 | |
| Disability level | | | | | |
| High§§ | 1.87 (1.45 to 2.43) | | | 1.88 (1.42 to 2.49) | <0.0001 |
| Low | 1.00 | | | 1.00 | |
| *Area-level variable* | | | | | |
| Townsend index | | | | | |
| Deprivation | | | | | |
| Yes (>0) | 1.74 (1.31 to 2.32) | 1.77 (1.33 to 2.36) | 1.65 (1.23 to 2.22) | 1.63 (1.21 to 2.21) | 0.001 |
| No (≤0) | 1.00 | 1.00 | 1.00 | 1.00 | |

*The parentheses show the reference categories that were used for each categorical variable when it was entered in the models—deprivation: [no] versus yes; GAD: [no] versus yes; education: [high] versus low; marital status: [married], single, others; social class: [non-manual] versus manual; lifetime MDD: [no] versus yes; prevalent physical disease: [no] versus yes; disability level: [low] versus high. These reference categories were based on the literature.[26 28–33] Choosing other groupings for the potential confounders would not have changed the results.
†Adjusted for age, SES (education, marital status, social class).
‡Adjusted for age, SES, lifetime MDD.
§Adjusted for age, SES, lifetime MDD, physical disease and disability.
¶High education: O-level, A-level, degree; low education: refers to no education.
**Other: divorced, separated, widowed.
††Manual: skilled manual, semiskilled, non-skilled; non-manual: professionals, managerial, skilled non-manual.
‡‡Prevalent physical disease: respiratory disease (asthma, bronchitis), allergies (allergies, hay fever), stroke, heart attack, cancer, diabetes, thyroid conditions, arthritis.
§§Below the median PCS value of 50.6.

in deprived circumstances.[23 48 49] Exposure to stress has been associated with central nervous system dysfunction and hypothalamic–pituitary–adrenal axis dysregulation, which have been implicated in the aetiology of GAD.[50 51] Women may also perceive the environment differently compared with men. Neighbourhood safety and fear of being sexually assaulted appear to be much more of a concern for women.[48 52] If women perceive their neighbourhood to be unsafe, they are less likely to engage in activities, such as walking, and this can negatively impact their mental health.[48 53] Perceiving neighbourhoods as unsafe can also erode social cohesion and can make women more hesitant to create social ties with others.[21] This can increase their risk of depression and

related mental disorders, because women derive health benefits from being embedded in social networks.[21] Living in deprivation can also make individuals feel excluded from society and ashamed,[54] and these feelings of exclusion are particularly harmful for women's mental health.[21 54]

Men and women may also perceive and exhibit the effects of stress in different ways.[55] Women who are highly distressed tend to develop internalising disorders, while men are more prone to substance abuse and anti-social personality.[56] The National Epidemiologic Survey on Alcohol and Related Conditions (NESARC) study[57] showed that total number of stressors experienced in life had a significantly stronger association with heavy

**Table 3** ORs for past-year generalised anxiety disorder according to individual-level and area-level characteristics for men (n=8219) who completed the Health and Life Experiences questionnaire in the European Prospective Investigation of Cancer-Norfolk cohort

| Characteristic* | ORs and 95% CI | | | | p Value for Model C |
|---|---|---|---|---|---|
| | Unadjusted | Model A† | Model B‡ | Model C§ | |
| *Individual-level variables* | | | | | |
| Sociodemographics | | | | | |
| Age (per 10 years) | 0.59 (0.49 to 0.71) | 0.58 (0.48 to 0.71) | 0.63 (0.51 to 0.77) | 0.52 (0.41 to 0.64) | <0.0001 |
| Education¶ | | | | | |
| Low | 0.91 (0.63 to 1.32) | 1.13 (0.75 to 1.70) | 1.16 (0.78 to 1.74) | 1.09 (0.73 to 1.63) | 0.670 |
| High | 1.00 | 1.00 | 1.00 | 1.00 | |
| Marital status | | | | | |
| Single | 2.65 (1.44 to 4.86) | 2.34 (1.26 to 4.36) | 2.67 (1.39 to 5.10) | 2.57 (1.32 to 5.01) | 0.006 |
| Married | 1.00 | 1.00 | 1.00 | 1.00 | |
| Other** | 2.09 (1.31 to 3.33) | 2.21 (1.39 to 3.52) | 1.48 (0.90 to 2.44) | 1.51 (0.91 to 2.51) | 0.111 |
| Social class†† | | | | | |
| Manual | 0.92 (0.65 to 1.29) | 0.83 (0.58 to 1.20) | 0.84 (0.58 to 1.23) | 0.74 (0.50 to 1.09) | 0.125 |
| Non-manual | 1.00 | 1.00 | 1.00 | 1.00 | |
| *Health status* | | | | | |
| Life-time MDD | | | | | |
| Yes | 15.38 (10.87 to 21.76) | | 14.25 (9.97 to 20.37) | 12.88 (8.99 to 18.46) | <0.0001 |
| No | 1.00 | | 1.00 | 1.00 | |
| Prevalent physical disease‡‡ | | | | | |
| Yes | 1.68 (1.20 to 2.35) | | | 1.53 (1.07 to 2.20) | 0.021 |
| No | 1.00 | | | 1.00 | |
| Disability level | | | | | |
| High§§ | 2.71 (1.88 to 3.90) | | | 3.10 (2.13 to 4.51) | <0.0001 |
| Low | 1.00 | | | 1.00 | |
| *Area-level variable* | | | | | |
| Townsend index | | | | | |
| Deprivation | | | | | |
| Yes (>0) | 1.36 (0.90 to 2.06) | 1.26 (0.82 to 1.94) | 1.19 (0.76 to 1.85) | 1.13 (0.72 to 1.77) | 0.598 |
| No (≤0) | 1.00 | 1.00 | 1.00 | 1.00 | |

*The parentheses show the reference categories that were used for each categorical variable when it was entered in the models—deprivation: [no] versus yes; GAD: [no] versus yes; education: [high] versus low; marital status: [married], single, others; social class: [non-manual] versus manual; lifetime MDD: [no] versus yes; prevalent physical disease: [no] versus yes; disability level: [low] versus high. These reference categories were based on the literature.[26] [28–33] Choosing other groupings for the potential confounders would not have changed the results.
†Adjusted for age, SES (education, marital status, social class).
‡Adjusted for age, SES, lifetime MDD.
§Adjusted for age, SES, lifetime MDD, physical diseases and disability.
¶High education: O-level, A-level, degree; low education: refers to no education.
**Other: divorced, separated, widowed.
††Manual: skilled manual, semiskilled, non-skilled; non-manual: professionals, managerial, skilled non-manual.
‡‡Prevalent physical disease: respiratory disease (asthma, bronchitis), allergies (allergies, hay fever), stroke, heart attack, cancer, diabetes, thyroid conditions, arthritis.
§§Below the median PCS value of 50.6.

drinking in men than in women. Therefore, men living in deprivation might be more likely to develop negative outcomes, such as heavy drinking, rather than anxiety.

### Strengths and weaknesses and future research

This study reveals that anxiety in women is strongly linked with area disadvantage. It has several strengths. We had a large, population-based sample of middle-aged and older-aged adults and adequately adjusted for a range of possible confounders. We used a structured, self-reported questionnaire to assess the presence of past-year GAD, and participants were followed for a long period of time. We overcome methodological limitations of previous studies by employing a commonly used,

theoretically sound measure of area deprivation capturing important features of the environment, such as unemployment and non-home ownership. We also had a large list of self-reported physician diagnoses of chronic physical diseases that we used to establish medical histories. Despite this, the residual effect of diseases not captured by our study, but that are associated with GAD may be present. Past illness may have been under-reported, which may have introduced measurement error and attenuated effect estimates towards the null. Participants were required to complete detailed dietary and lifestyle questionnaires and undergo periodic health assessments. Since those who participated in EPIC-Norfolk were somewhat less deprived and healthier

than individuals living in other parts of England,[25] [31] our results may not generalise to people living in extremely deprived circumstances. When comparing the demographic characteristics of responders versus non-responders (see online supplementary appendix I), we found that participants were slightly younger and slightly more women than men consented. The association found within our cohort is unlikely to be explained by selection bias. It is unlikely that the association in non-responders would be in the opposite direction to that which we obtained in our study.

Another limitation is that some of the areas classified as deprived in 1991 might have shown an improvement in socioeconomic circumstances over time and become more affluent, and vice versa. Although this might present an issue for samples drawn from busy, urban environments, we expect changes in area-level circumstances for the EPIC-Norfolk cohort to have been small. Many EPIC-Norfolk participants come from rural areas, where significant urban development and change in the residential environment are unlikely to have occurred during the study period.[31] Nonetheless, to account for potential changes in GAD rates and area-level circumstances, future studies should assess the association between anxiety and area deprivation at multiple time points.

Although area deprivation was measured in 1991 and GAD in 1996–2000, we expect the association between anxiety and area deprivation in women to be even stronger with more recent data. First, older, as well as, more recent literature has shown that poor women or those living in disadvantage are more likely to develop negative health outcomes, while men less so.[23] [46] [58] Second, women are increasingly taking on multiple roles in society, such as income-earner, childbearer and carer, which is adding to their burden (especially if they are living in deprivation).[24] Third, research has also shown that anxiety rates have been increasing in women in recent times.[59] For these reasons, we expect the association between area deprivation and GAD to be even stronger in women at the present time.

Future research should consider assessing the risk of GAD in countries with high social and material inequalities, such as the USA, where the rates of anxiety are also some of the highest in the world.[60] Compared with the UK, the overall prevalence of GAD in the USA is more than twice as high, and middle-aged people are most affected, with a prevalence of 7.7%.[61] It would be especially informative to repeat this study in less developed parts of the world, such as India, where poverty is strongly linked to the development of mental disorders, and women's unequal status and social roles in society represent important additional issues.[62]

### Implications for future generations and placing our research in context

The consequences of living in deprivation are far-reaching and can affect future generations. Repeated exposure to socioeconomic disadvantage in childhood is a consistent predictor of poor mental health in adolescence and young adulthood, particularly for young girls.[63] Since anxiety disorders tend to emerge in early adolescence, repeated exposure to socioeconomic disadvantage in childhood can increase the risk for more severe, early-onset forms of the disorder. Early-onset forms are the most difficult to treat and have a poor prognosis.[64] Our study is the largest to date to examine the link between area deprivation and GAD.

Although other studies have shown that the places where people live have a substantial impact on health,[15] [16] studies on the links between area deprivation and mental disorders among men and women, separately are limited. A recent, large, population-based study[18] of over 21 000 people living in Ireland showed that area deprivation was associated with a significantly increased risk for common mental disorders in women, but not in men, after controlling for demographic and socioeconomic factors. In line with this, a study[21] of over 2700 adults living in Canada showed that greater neighbourhood disadvantage also was associated with increased risk of depressive symptoms in women, but not in men. Research conducted in the USA had similar findings.[54] This indicates that characteristics of the living context seem to influence women's health in particular. Very few studies have assessed the association between deprivation and mental health among women and men, separately and research specifically focusing on anxiety disorders is scarcer still.

Our findings differ from the only other population-based, contextual study of generalised anxiety among men and women living in areas of low socioeconomic circumstances.[23] In this cross-sectional study, no association with anxiety was found; however, the measure of deprivation was based only on the local unemployment rate and median area income. Thus, the results are not directly comparable to ours. Further, the previous study used the Symptoms Checklist-90-Revised scale to measure *symptoms* of generalised anxiety, yielding different estimates than ours. In contrast to the DSM-IV, the Symptoms Checklist-90-Revised scale did not base the definition of generalised anxiety around excessive, uncontrollable worry, which is the central, defining feature of GAD, and used a much shorter time frame to assess symptoms. We used a thorough assessment of DSM-IV GAD, which was measured in the past year. In contrast to the previous study, we also examined area deprivation using a common, theoretically-sound index, covering a wide range of key domains relating to socioeconomic disadvantage, such as non-home ownership and non-car ownership. Studies assessing other health outcomes have suggested that the residential environment has a larger effect on women's health,[23] [46] while individual-level factors relating to social status, such as employment, have the greatest impact on men's health.[45] Among disadvantaged women, it is not lack of money per se that leads to poorer health, but rather the inability to derive the necessary resources from the

environment to make ends meet; this can translate into stress and anxiety.[65] Women are becoming financially independent as they enter the labour force, which means that economic hardship now impacts them, as well. Women perceive economic hardship as a barrier to managing daily life and making ends meet, which can increase their anxiety. In contrast, men link joss loss to a decline in social status.[45 57 65] When men experience job-related stresses, they tend to externalise the effects of such stress and develop substance abuse.[57 59]

## Interpretation

The absolute numbers of people living in deprived conditions are large worldwide. This, combined with a growing mental health burden, means that the findings obtained in this study remain highly relevant. The WHO[66] has emphasised the need to reduce social and health inequalities. Our findings provide a strong evidence base to this call, showing that 'perhaps the most important risks to health are beyond people's immediate control'[67] and that the environment needs to be taken into account when developing mental health policy. Gender is important when it comes to assessing the impacts of the environment on our mental health. Our study shows that investments made to improve local areas will not impact men and women in the same way. Regarding clinical implications, health professionals should consider assessing anxiety in women living in deprivation.

**Acknowledgements** OR received a PhD studentship from the National Institute for Health Research.

**Contributors** OR (corresponding author) had the idea for and conducted the analysis, and wrote the article, along with CB, K-TK, LL, PS and NW. All authors provided feedback into the analysis and critically reviewed drafts of the manuscript. All authors have seen and approved the final version. The authors had full access to all the data in the study and take responsibility for the integrity of the data and the accuracy of the data analysis. OR acts as the guarantor of the study.

**Funding** This work was supported by the Medical Research Council UK (grant number G9502233) and Cancer Research UK (grant number SP2024-0201 and SP2024-0204).

**Competing interests** None declared.

**Ethics approval** The study has ethics committee approval from Norfolk Ethics Committee (Rec Ref: 98CN01).

**Provenance and peer review** Not commissioned; externally peer reviewed.

**Data sharing statement** No additional data are available.

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
