## [Reviewer comments · BMJ Open]

ARTICLE DETAILS

TITLE (PROVISIONAL)	Sex differences in the association between area deprivation and generalised anxiety disorder: British population study
AUTHORS	Remes, Olivia; Wainwright, Nicholas; Surtees, Paul; LaFortune, Louise; Khaw, Kay-Tee; Brayne, Carol

VERSION 1 - REVIEW

REVIEWER	Martina Behanova Faculty of Medical Sciences Antonius Deusinglaan 1 9713 AV Groningen The Netherlands
REVIEW RETURNED	09-Oct-2016

GENERAL COMMENTS	This is very nicely written paper. The introduction is clear and concise, the method is well elaborated and the statistical methods seem in order, the result is presented in a pleasant way, and the discussion is well balanced. A strong point of this study is that it used data from the EPIC-Norfolk study, a large, UK population based study on more than 30 000 respondents. It aimed to assess the effect of area deprivation on generalised anxiety disorder (GAD) and whether this differs between men and women. Authors used reliable and validated measures for GAD as well as for area deprivation. Even though, I have some comments that may be considered for fine tuning of this manuscript. General comments: You used two indicators of area deprivation, but you didn't provide any rationale for choosing them. In results, you present outcomes for the Townsend index, however in a Discussion part you discuss domains which are covered by the Index of Multiple deprivation indicator. Further, results for the Index of Multiple deprivation are not presented correctly. More consistency and clarity on this would be helpful. Many times you state that you did something according to the literature but you don't cite any. Please do so. Introduction: - You may support and extend your hypotheses by recalling studies which shown the independent effect of area deprivation on depression, anxiety or mental health problems (in general). For example see: Matheson FI, Moineddin R, Dunn JR, Creatore MI, Gozdyra P, Glazier RH. Urban neighborhoods, chronic stress, gender and depression. Soc Sci Med 2006; 63:2604-2616.
---

	Galea S, Ahern J, Nandi A, Tracy M, Beard J, Vlahov D. Urban neighborhood poverty and the incidence of depression in a population based cohort study. Ann Epidemiol 2007; 17:171-179.  - Line 101: I would mention here the Gini coefficient for UK, a measure for inequality. - I would recommend you to put a sentence on a line 107-111 to a methods part. Next, please provide your rationale and cite relevant literature for doing analysis separately for men and women. Methods:  - Line 150: you do not provide information on a final categorization for social class, that is, you merged it into 2 categories only, manual vs.non-manual. - Line 158: Why did you merged education into two categories only? - Line 171-172: Please cite here relevant literature. - What was the rationale behind choosing the particular two indicators of area deprivation? Why did you prefer to use the Townsend index instead of the Carstairs index? The Carstairs index comprises also a proportion of low social class whereas the Townsend index not. Then, why did you choose to use data from 1991 for the Townsend index and from 2001 for the Index of Multiple deprivation? I miss these information here. - Line 207-208: You don't provide overview on the distribution of respondents per IMD score quartile. Please do so in Table 1. - Line 211: (...).were constructed THEN adjusted for... - Line 217: If most of the analysis used the Townsend index as a measure of area deprivation, why did you choose the second indicator? - Line 219-223: I would completely omit this text from Methods and would rather put it in footnotes under table(s). - Line 225-226: Please provide relevant literature sources. RESULTS:  - Table 1: I recommend you to slightly modify a Table by adding a heading for variables concerning individual-level data and then for area-level variables. Please note that IMD index is now missing in Table 1. - Line 269: I recommend you to change "more likely to be younger" into "were less than 50 years old". - Line 271: Please add a reference to your Table 1. - Table 2: I recommend you to modify also a Table 2. A column with Model C is now misleading. It should show results for quartiles of IMD scores but the table doesn't have any rows for it. It will be useful if you add both measures of area deprivation in a Table. Discussion: Line 336-337: I think that this reference is not relevant here as it refers to adolescent girls and your study population is above 40 years and the effect of area deprivation on health is different for this two groups.
--	---

REVIEWER	Ellen Generaal, PhD VU University Medical Center, Dept. of Psychiatry, Amsterdam, The Netherlands
REVIEW RETURNED	20-Oct-2016

GENERAL COMMENTS	This is a large-scale population study among 10275 women and
--

8219 men in England aged 40 years and older. The role of area deprivation (e.g. unemployment) is examined in relation to generalized anxiety disorders (GAD), while adjusting for a number of individual and health-related confounders. The authors are to be complimented on the large sample size and the novelty of the topic. However, there are several major concerns regarding the research design and the selection of the independent variable, which limit the potential contribution to the literature:

1) The timepoints at which the authors measured the independent and dependent variable are different. The authors measured area deprivation in 1991 and 2001, whereas GAD was measured between 1996-2000. Thus, it is possible that their findings concern associations between area deprivation and a diagnosis of GAD occurring 1 to 9 years later. This worries me as a reader, in particular because factors such as unemployment and household overcrowding change constantly over time. Also, in case of GAD, despite low remission rates, a person can show remission over time (Yonkers et al., Depr&Anx, 2003). Thus, differences in the chosen timepoints clearly limit the authors in drawing conclusions from their findings. This issue is not even raised in the Discussion section.

2) It is unclear to me how the authors came to the score for the Townsend index for area deprivation. "Townsend index scores were used to create a dichotomous variable, with 0 as the cut-point (representing the national average)" (lines 206-208). The authors refer to an earlier study explaining this index. This explanation should be included in this paper to prevent readers having to go back and forward. Also, I am doubtful about using area deprivation as a dichotomous measure (yes/no). It may be more informative to categorize subjects according to their level of deprivation (1-4 environmental risk factors), and to indicate in the results section which components of the deprivation score (e.g. unemployment, non-car ownership) are most strongly related to GAD. This would result in more specific information regarding which type of deprivation might be associated with GAD.

Minor issues:

- The definition of area deprivation is not explained until the method section. Please clarify what is meant by 'area deprivation' in the abstract and the introduction: '(e.g. income, employment)'.

- Abstract: 'individual-level' circumstances, please provide an example.

- Strengths & Limitations bullet points:

* 'a range of important confounders' (line 74) Please provide an example to clarify what the authors consider to be 'important' confounders.

* "for a long period of time" (line 77). Please indicate how long.

Introduction

- line 111, please write 'area deprivation'.

Methods

- line 125, please explain 'During follow-up' (what years?)

- line 177, please explain the abbreviation IMD when first mentioned in the manuscript "Index of Multiple Deprivation (IMD)"

- Statistical analyses: please include which statistical program was used for the regression analyses.

- line 225, what is meant by "It was not possible to group the GAD

	variable otherwise...” - line 228(+317), please replace ‘pure GAD’ by ‘GAD without MDD’ (or something similar) Results - line 291, please indicate p-value and remove ‘highly’. The interpretation of the strength of the association is up the reader by looking at the OR, CI and p-value. Discussion - line 332, please explain what is meant by ‘rigorous’ analyses - line 336 “.. and poor self-related health” Please add reference. - line 355, “..of a concern to women”. Please add reference. - lines 362-375. Please make sure the reader understands that the role of air pollution in the relation between deprivation and anxiety is based on previous studies. It now seems that pollution could have been measured in the current study.
--	---

REVIEWER	Ann John Swansea University Medical School
REVIEW RETURNED	04-Nov-2016

GENERAL COMMENTS	The data used in this study is very old. There is no discussion of this and how relationships between gender and deprivation may have changed since the late 90s / 2001 IMD is not adequately described p8, l178- it is not based on Census data but includes some Census data Why categorise IMD to quartiles not quintiles as is standard? Original population 77,630; 30445 consented: 20,921 completed follow up HLEQ; 18494 eligible for analysis- 18494/30445+60% follow up so potential for selection bias. Would need a demographic table to see if those responding different from original cohort. No mention of this selection bias in the limitations section. The literature on mechanisms for the association in women and deprivation described here is extensive in relation to Common Mental disorders and should be quoted more rather than the focus on environmental issues (ie educational attainment. There should be a lot more discussion of the negative finding in males- this is possibly out of keeping with what you would expect. Why this method and not multi level analysis?
---

VERSION 1 – AUTHOR RESPONSE

Reviewer 1: Dr. Martina Behanova

1. Comment:

This is very nicely written paper. The introduction is clear and concise, the method is well elaborated and the statistical methods seem in order, the result is presented in a pleasant way, and the discussion is well balanced. A strong point of this study is that it used data from the EPIC-Norfolk study, a large, UK population based study on more than 30 000 respondents. It aimed to assess the effect of area deprivation on generalised anxiety disorder (GAD) and whether this differs between men and women. Authors used reliable and validated measures for GAD as well as for area deprivation.

Response:

We would like to thank Dr. Behanova for her positive comments highlighting the large size of the study, and reliable, validated measures of GAD and area deprivation.

Also, thank you very much for mentioning that our paper is clear and well-written, and that the results and discussion are presented well. We have addressed all the comments you provided and made all the changes suggested.

This is, indeed, the first paper to assess the association between deprivation and generalized anxiety disorder in men and women, and we believe the findings are important for clinicians and policy-makers.

2. Comment:

You used two indicators of area deprivation, but you didn't provide any rationale for choosing them. In results, you present outcomes for the Townsend index, however in a Discussion part you discuss domains which are covered by the Index of Multiple deprivation indicator. Further, results for the Index of Multiple deprivation are not presented correctly. More consistency and clarity on this would be helpful.

Many times you state that you did something according to the literature but you don't cite any. Please do so.

Response:

Thank you very much for these comments. Our rationale for using the Townsend index is that it is one of the most commonly-used measures of area deprivation, it is theoretically sound, and captures important features of the environment. We have indicated in the Methods, that in order "to examine area deprivation, we used one of the most commonly-used measures". And in the discussion we stated that we employed "a commonly-used, theoretically-sound measure of area deprivation capturing important features of the environment, such as, unemployment and non-home ownership". Also other studies assessing deprivation have used this index, including publications based on the EPIC-Norfolk cohort (ex. Shohaimi 2003).

To avoid confusion (and also as per reviewer 2's comments), we have removed the analyses based on the Index of Multiple Deprivation. We agree that this improves clarity and consistency in the presentation of our results.

We have also provided more citations throughout our text, especially when we said that we did something according to the literature, as requested.

--Comments pertaining to Introduction section--

3. Comment:

You may support and extend your hypotheses by recalling studies which shown the independent effect of area deprivation on depression, anxiety or mental health problems (in general). For example see:

Matheson FI, Moineddin R, Dunn JR, Creatore MI, Gozdyra P, Glazier RH. Urban neighborhoods, chronic stress, gender and depression. *Soc Sci Med* 2006; 63:2604-2616.

Galea S, Ahern J, Nandi A, Tracy M, Beard J, Vlahov D. Urban neighborhood poverty and the incidence of depression in a population based cohort study. *Ann Epidemiol* 2007; 17:171-179.

Response:

Thank you for providing us with these references. We have now included findings from these studies,

in the Introduction (references 19 and 20). We were unable to elaborate on this further because of word space considerations.

This is the section we have added to the Introduction:

“Population-based studies further showed that living in disadvantaged neighbourhoods or places where there is high chronic stress can increase the risk for mental disorders, such as, depression.[18, 19, 20]. Whether area deprivation can be used to predict generalized anxiety disorder (GAD) is unknown.”

4. Comment:

Line 101: I would mention here the Gini coefficient for UK, a measure for inequality.

Response:

We included literature on this and provided the reference for it, as requested.

This is the section we included in the Introduction:

“Many studies conducted in western countries have shown that living in areas characterized by high income inequality can lead to significantly increased risks for serious medical conditions and mortality.[15, 16] A meta-analysis of cohort studies including over 60 million participants showed that people living in areas of high income inequality, as measured by the GINI index, had an increased risk for mortality.[17] ”

5. Comment:

I would recommend you to put a sentence on a line 107-111 to a methods part. Next, please provide your rationale and cite relevant literature for doing analysis separately for men and women.

Response:

We moved the section to the Methods, as requested.

As indicated, we provided a comprehensive rationale for doing analyses separately for men and women. We have now included a section on this in the Introduction and cited the relevant literature for this.

This is the rationale that we included for doing analyses separately for men and women:

“Results are presented separately for women and men, and this is done for several reasons. Research has shown that women are more likely to develop anxiety compared to men, mainly due to biological, social, and environmental factors.[21, 22, 23] Gender frames access to resources derived from the environment.[21, 22] Compared to men, women have been shown to have less access to material and social conditions, and this can influence mental health. Women also seem to interact with their environment differently. For example, women are exposed to different types of stressors compared to men, because of gender differences in social roles.[18, 21]

Despite these differences, research examining the link between the living context, such as, area deprivation, and mental health from a gendered perspective is scarce. It remains unclear whether there are sex differences in the association between area deprivation and risk of generalized anxiety disorder – and our objective in this study will be to assess this. Knowing that one sex is at risk of developing anxiety when exposed to deprived circumstances helps to tailor interventions and allocate scarce resources according to need.[24]”

--Comments pertaining to Methods and Results sections--

6. Comment:

Line 150: you do not provide information on a final categorization for social class, that is, you merged

it into 2 categories only, manual vs. non-manual.

Response:

Thank you very much for this. We have now provided additional information on this: "The final categorization of social class included manual: skilled manual, partly skilled, and unskilled; and non-manual: professionals, managerial and technical, and skilled non-manual."

7. Comment:

Line 158: Why did you merged education into two categories only?

Response:

We merged education into two categories in order to have enough people with GAD in each category (cell size considerations). Also, these categories were used in other EPIC-Norfolk publications. We have now specified this in the Methods: "The final categorization of the variables took cell size into account and was also done in accordance with previous literature.[26, 28-33]."

8. Comment:

Line 171-172: Please cite here relevant literature.

Response:

We have now done so, as requested.

9. Comment:

What was the rationale behind choosing the particular two indicators of area deprivation? Why did you prefer to use the Townsend index instead of the Carstairs index? The Carstairs index comprises also a proportion of low social class whereas the Townsend index not. Then, why did you choose to use data from 1991 for the Townsend index and from 2001 for the Index of Multiple deprivation? I miss these information here.

Response:

Thank you very much for this comment.

This was our rationale: the Townsend index is a commonly-used measure of area deprivation in the UK and it is theoretically-sound.

Also, in a letter sent to the BMJ (BMJ 1991) by the researchers who developed the Carstairs index and the Townsend index, it is mentioned that both of these indices (Carstairs and the Townsend) are very similar and both are based on a combination of several socioeconomic variables.

These indices were both developed as measures of material deprivation. They also correlate strongly with similar outcomes, such as, mortality. (Carstairs 1991)

According to research (Adams 2004), the "Townsend index is a commonly-used and theoretically sound measure of socio-economic position with a clearly stated rationale." In a study conducted to measure the ability of different measures of socio-economic position to predict self-reported health, the Townsend index fared just as well as other measures. (Adams 2004)

Finally, a number of studies assessing deprivation have used the Townsend index (ex. British studies: Shohaimi 2014, Cooper 2012, Shohaimi 2003; international studies: Bertin 2014, Lopez de Fede 2016), including studies based on the EPIC-Norfolk cohort.

10. Comment:

Line 207-208: You don't provide overview on the distribution of respondents per IMD score quartile.

Please do so in Table 1.

Response:

We have simplified the analysis by removing the IMD and focussing solely on the Townsend index. This was also done in relation with reviewer 2's comments.

11. Comment:

Line 211: (..)were constructed THEN adjusted for...

Response:

We agree that our wording is slightly ambiguous. We have changed this, but prefer "were constructed to...", rather than "constructed then...".

12. Comment:

Line 217: If most of the analysis used the Townsend index as a measure of area deprivation, why did you choose the second indicator?

Response:

We agree. We have now removed the IMD and focused the paper solely on the Townsend index to improve clarity and consistency in the presentation of our results.

Our rationale for having included the IMD was that it is another frequently-used measure of area deprivation. We wanted to show that both of these measures of deprivation show highly significant associations with anxiety in women, and not in men – and they did, which gave us great confidence in our findings. However, we agree that it is better to focus the paper solely on the Townsend index.

13. Comment:

Line 219-223: I would completely omit this text from Methods and would rather put it in footnotes under table(s).

Response:

We have now placed the text in the footnotes under tables 2 and 3, which include the results from the multivariable models. The reference categories listed were used for the categorical variables when they were entered in the multivariable models (the results of these models are provided in tables 2 and 3).

14. Comment:

Line 225-226: Please provide relevant literature sources.

Response:

We have now cited the literature.

15. Comment:

Table 1: I recommend you to slightly modify a Table by adding a heading for variables concerning individual-level data and then for area-level variables. Please note that IMD index is now missing in Table 1.

Response:

We have now added in "individual-level" and "area-level" variable headings to table 1, as requested. For consistency, we have also done this to tables 2 and 3.

As per reviewer 2's comments, we have removed the analyses on the IMD.

16. Comment:

Line 269: I recommend you to change “more likely to be younger” into “were less than 50 years old”.

Response:

We have made this change, as requested.

17. Comment:

Line 271: Please add a reference to your Table 1.

Response:

We have now provided the citation.

18. Comment:

Table 2: I recommend you to modify also a Table 2. A column with Model C is now misleading. It should show results for quartiles of IMD scores but the table doesn't have any rows for it. It will be useful if you add both measures of area deprivation in a Table.

Response:

Thank you very much for this comment. As per reviewer 2's suggestions, we have now removed the analyses pertaining to the IMD and focused the paper solely on the Townsend index to improve clarity and consistency in the presentation of our results.

--Comment pertaining to Discussion section--

19. Comment:

Line 336-337: I think that this reference is not relevant here as it refers to adolescent girls and your study population is above 40 years and the effect of area deprivation on health is different for this two groups.

Response:

We have now removed this reference, as requested. We have slightly altered this paragraph to include suggestions made by reviewer 2.

Reviewer 2: Dr. Ellen Generaal

1. Comment:

This is a large-scale population study among 10275 women and 8219 men in England aged 40 years and older. The role of area deprivation (e.g. unemployment) is examined in relation to generalized anxiety disorders (GAD), while adjusting for a number of individual and health-related confounders. The authors are to be complimented on the large sample size and the novelty of the topic. However, there are several major concerns regarding the research design and the selection of the independent variable, which limit the potential contribution to the literature:

1) The timepoints at which the authors measured the independent and dependent variable are different. The authors measured area deprivation in 1991 and 2001, whereas GAD was measured between 1996-2000. Thus, it is possible that their findings concern associations between area deprivation and a diagnosis of GAD occurring 1 to 9 years later. This worries me as a reader, in

particular because factors such as unemployment and household overcrowding change constantly over time. Also, in case of GAD, despite low remission rates, a person can show remission over time (Yonkers et al., Depr&Anx, 2003). Thus, differences in the chosen timepoints clearly limit the authors in drawing conclusions from their findings. This issue is not even raised in the Discussion section.

Response:

Thank you very much for the positive comments highlighting the large sample size of this study, the novelty of this topic, and the fact that we adjusted for a number of individual and health-related confounders.

We have carefully gone through each comment you provided us with and made the changes, as requested. We would like to thank you for taking the time to go through our manuscript and for the detailed feedback you provided us with – we have incorporated the comments and improved our paper.

Regarding the comment on anxiety being measured in 1996-2000 and the IMD in 2001 - we agree that this is slightly confusing. To improve clarity and consistency in the presentation of our results, we have now removed the IMD (which was measured slightly after anxiety was assessed) and focused the paper solely on the Townsend index (which was measured in 1991, before anxiety was measured in 1996-2000).

Our rationale for having included the IMD was that it is another frequently-used measure of area deprivation – it gave us similar results to the Townsend (and therefore, gave us great confidence in our findings). Nevertheless, we agree that it is better to simply the analysis and focus the paper solely on the Townsend index, which is measured before anxiety. The inclusion of the IMD had also raised some questions for reviewer 1.

The other comment you made (relating to anxiety levels and deprivation changing over time):

Thank you very much for pointing out that we needed to include a section on this in the Discussion. We have now added in the following:

“Another limitation is that some of the areas classified as deprived in 1991 might have shown an improvement in socioeconomic circumstances over time and become more affluent, and vice versa. Although this might present an issue for samples drawn from busy, urban environments, we expect changes in area-level circumstances for the EPIC-Norfolk cohort to have been small. Many EPIC-Norfolk participants come from rural areas, where significant urban development and change in the residential environment are unlikely to have occurred during the study period.[31] Nonetheless, to account for potential changes in GAD rates and area-level circumstances, future studies should assess the association between anxiety and area deprivation at multiple time points.”

As indicated in the previous passage, we do not expect the link between area deprivation and anxiety in women to have changed over time for a few reasons. Many EPIC-Norfolk participants come from rural areas, where urban development and change in residential circumstances is less likely to have occurred during the study period and over the length of time we looked at. There is also very little outward migration in the EPIC-Norfolk study region, and recent research also indicates that GAD tends to have a chronic and persistent course (ex. Zhang et al. 2015 *Transl Psychiatry*, Hoge et al. 2012 *Br Med J*). Nevertheless, it would be useful to model the change in area deprivation and the rate of change in anxiety over time (thus, probing the dynamic influence of area deprivation on anxiety in residents). We were unable to measure area deprivation at a later time point, because the Census is conducted every 10 years – and including information from the 2001 Census would have meant that area deprivation would have been measured after anxiety was assessed in this study.

Of interest, one of the few longitudinal studies modeling the dynamic influence of neighbourhood

unemployment on depressive symptoms in middle-aged and older adults (Wight 2013) showed the following: a neighbourhood's unemployment history is associated with depression a decade later among residents, independent of the neighbourhood's current unemployment level. This shows that people exposed to deprivation at one point in time, even 10 years before mental disorders are measured, take an 'emotional hit' from living in disadvantage.

Nevertheless, as you have pointed out, this study provides novel results. We also had access to a very large sample, took into account a range of important confounders (in accordance with the literature), used a structured questionnaire to derive a DSM-based measure of GAD, and used information from the Census to derive a measure of deprivation levels (avoiding the same-source bias that is present in many studies).

2. Comment:

It is unclear to me how the authors came to the score for the Townsend index for area deprivation. "Townsend index scores were used to create a dichotomous variable, with 0 as the cut-point (representing the national average)" (lines 206-208). The authors refer to an earlier study explaining this index. This explanation should be included in this paper to prevent readers having to go back and forward. Also, I am doubtful about using area deprivation as a dichotomous measure (yes/no). It may be more informative to categorize subjects according to their level of deprivation (1-4 environmental risk factors), and to indicate in the results section which components of the deprivation score (e.g. unemployment, non-car ownership) are most strongly related to GAD. This would result in more specific information regarding which type of deprivation might be associated with GAD.

Response:

Thank you for pointing out that we needed to better explain how we came to the score for the Townsend index. We have now included a longer and more detailed section on how the index was constructed – under the heading "Area-level measure (exposure variable)" in the Methods.

This is the more detailed, clearer section on the index:

"To examine area deprivation, we used one of the most commonly-used measures of area deprivation in the UK: the Townsend Index[41, 42]. This index is a composite measure of four variables obtained from the 1991 Census: 1) percentage of economically active residents over age 16 who are unemployed, 2) percentage of households that do not possess a car, 3) percentage of private households that are not owner occupied, and 4) percentage of private households that are overcrowded (have more than 1 person per room). These variables were obtained at the level of the enumeration district. Each variable was standardized by obtaining Z scores (dividing the mean by the standard deviation across enumeration districts in England). The Z-values of the four variables were added together to produce a Townsend index score for each enumeration district. Positive values of the index indicate enumeration districts that are more deprived, while negative values indicate those that are less deprived; 0 represents the national mean. The postal codes of participants were recorded linked to enumeration districts, and participants were considered to live in deprived areas depending on the Townsend index score assigned to their enumeration district.[41]

The Townsend deprivation index was also disaggregated into its four constituent components to determine whether any one of these is associated with GAD or if it is the effect of the combined components that is important."

In line with what other studies had done (Shohaimi 2003, Shohaimi 2004, Surtees 2004), we dichotomised the Townsend index scores at 0, because this value (0) represents the national average, and allows comparison of those who are more deprived (above this cut-point) with those who are less deprived (below 0).

We have also disaggregated the Townsend index and re-ran the fully-adjusted model with the four

constituent components of the Townsend index, as you had requested. We have updated our Methods, Results, and Discussion sections, accordingly.

3. Comment:

The definition of area deprivation is not explained until the method section. Please clarify what is meant by 'area deprivation' in the abstract and the introduction: '(e.g. income, employment)'.

Response:

Thank you very much for this. We have now clarified what we mean by area deprivation in the abstract and in the introduction – we provided examples, as suggested.

In the abstract we added in: "Studies have shown that area-level deprivation measured by factors, such as, non-home ownership, non-car ownership, and household overcrowding..."

In the introduction we added in:

"Area deprivation refers to residential environments or living contexts characterized by factors, such as, high levels of unemployment, non-home ownership, non-car ownership, and low income.[14]"

4. Comment:

Abstract: 'individual-level' circumstances, please provide an example.

Response:

We have now provided an example, as requested.

We have added in:

"...individual-level circumstances, such as, education and social class."

5. Comment:

- Strengths & Limitations bullet points:

* 'a range of important confounders' (line 74) Please provide an example to clarify what the authors consider to be 'important' confounders.

Response:

We have now provided examples to clarify what we mean by important confounders, as requested.

We have added in:

"We used a large, population-based sample of middle- and older-aged adults and adjusted for a range of important confounders, such as, sociodemographic factors and medical history."

6. Comment:

* "for a long period of time" (line 77). Please indicate how long.

Response:

We have now indicated the length of time.

--Comments pertaining to Introduction and Methods sections--

7. Comment:

- line 111, please write 'area deprivation'.

Response:

As per comments from reviewers, some of the text was slightly rearranged. However, we made sure to write 'area deprivation' where relevant, as requested (for example, in the Introduction: "Whether

area deprivation can be used to predict generalized anxiety disorder (GAD) is unknown”).

8. Comment:

- line 125, please explain ‘During follow-up’ (what years?)

Response:

We have now clarified this.

9. Comment:

- line 177, please explain the abbreviation IMD when first mentioned in the manuscript “Index of Multiple Deprivation (IMD)”

Response:

Because of the confusion the IMD was causing to reviewers, we agreed that it is better to remove it and focus solely on the Townsend index.

10. Comment:

- Statistical analyses: please include which statistical program was used for the regression analyses.

Response:

Thank you for noting this. We had previously mentioned that analyses were implemented in SAS Version 9.3 (SAS Institute, Cary, NC)” - others papers using SAS programs for their regressions had written similar statements. We have now provided the full name for SAS for additional clarity – Statistical Analysis Software.

11. Comment:

- line 225, what is meant by “It was not possible to group the GAD variable otherwise...”

Response:

We have now clarified this statement.

When the HLEQ questionnaire was administered in 1996-2000, psychosocial information was collected so that a dichotomous GAD variable could be created in accordance with DSM-IV; therefore, it was not possible to re-group the anxiety symptoms otherwise (other than in accordance with DSM-IV, because this was the initial intention when the questionnaire was designed). As per your comments, we have clarified our statement, and added in: “since it was created and categorized according to the DSM-IV definition”.

12. Comment:

- line 228(+317), please replace ‘pure GAD’ by ‘GAD without MDD’ (or something similar)

Response:

We made the change; we wrote GAD without MDD.

--Comments pertaining to Results section--

13. Comment:

- line 291, please indicate p-value and remove ‘highly’. The interpretation of the strength of the association is up the reader by looking at the OR, CI and p-value.

Response:

We have now removed 'highly' and included the p-value, as requested.

--Comments pertaining to Discussion section--

14. Comment:

- line 332, please explain what is meant by 'rigorous' analyses

Response:

We have clarified this section and added in the following:

"It is difficult to confirm causality between area deprivation and GAD; however, a rigorous analysis based on observational data is a reasonable method of examining this relationship. The analysis was rigorous, because we used reliable and commonly-used measures of area deprivation and GAD, controlled for covariates that are associated with the exposure (area deprivation) and outcome (GAD), had access to a large sample size of over 18,000 people, and followed participants for a long period (7 years)."

15. Comment:

- line 336 ".. and poor self-related health" Please add reference.

Response:

We have refined this section in accordance with comments from reviewers. This sentence is no longer here - we elaborated on previous research assessing the link between area deprivation/socioeconomic disadvantage and mental health in women in a different part of the Discussion section. However, we made sure to properly reference any literature discussed in the paper.

16. Comment:

- line 355, "..of a concern to women". Please add reference.

Response:

We have now provided the citation.

17. Comment:

- lines 362-375. Please make sure the reader understands that the role of air pollution in the relation between deprivation and anxiety is based on previous studies. It now seems that pollution could have been measured in the current study.

Response:

Thank you for pointing this out. We agree that it is somewhat confusing to mention pollution, because we did not conduct such analyses. We have therefore omitted this section and have expanded the section summarizing the links between gender, deprivation, and mental disorders (in line with reviewer 3's comments).

Reviewer 3: Dr. Ann John

1. Comment:

The data used in this study is very old. There is no discussion of this and how relationships between gender and deprivation may have changed since the late 90s / 2001

Response:

Thank you very much for the comments you provided us with – we have used them to improve our manuscript. We went over each comment carefully and made as many changes as possible. We reviewed the literature again and clarified sections which needed additional explanations, and ran additional analyses, as requested.

As indicated, we have now added in a section in the Discussion mentioning how relationships between gender and deprivation may have changed since that time. We agree that a discussion on this is warranted.

Older and more recent studies (ex. Pattyn 2011, Morrissey 2016) suggest that disadvantage leads to poor health outcomes, such as, depression, in women, but not men. Therefore, we would also expect the association between deprivation and anxiety in women to hold over time. At the present time, the World Health Organization also indicates that socioeconomic disadvantage is a “gender specific risk factor for common mental disorders”.

However, three other reasons suggest that the association between deprivation and anxiety should be even stronger today compared to 10 or 20 years ago. A Swedish study (Ahnquist 2007) showed that women who had been exposed to financial stress during their lives were at a high risk for negative health outcomes. Even though men have traditionally had the responsibility of being the breadwinner, women in western countries have increasingly entered the labour force over the past couple of decades and are becoming financially independent. This means that economic hardship now impacts women, as well. The Swedish study (Ahnquist 2007) showed that economic hardship has been having an increasingly detrimental effect on women’s health over time (evidenced by a dose-response association between exposure to financial stress and negative health in women; this was not seen in men). For women, however, it is not low income in itself that leads to poorer health, but rather the inability to make ends meet on a day-to-day basis. (Ahnquist 2007) This inability to derive the necessary resources from their environment and manage daily life can translate into increased anxiety. Second, women are increasingly taking on multiple roles in society, such as, carer, income-earner, and child-bearer, which increases the burden and stress placed on this sex group (Remes et al. *Lancet Psychiatry* in press 2017). Third, there is emerging evidence that anxiety rates in women have been increasing in recent times, as well (NHS Digital Report 2016). For these reasons, we believe that we should be seeing an even stronger association between deprivation and anxiety in women at the present time. Regarding men: recent studies still support the hypothesis that deprivation increases poor health outcomes in women, and less so in men. A study published very recently (Morrissey 2016) was in line with our findings – it showed that women living in deprivation were at increased risk for common mental disorders, but not men. This study was frequently cited in our paper.

We could not include all the aforementioned details and we had to be concise, because of word count considerations. This is what we added to the paper:

“Although area deprivation was measured in 1991 and GAD in 1996-2000, we expect the association between anxiety and area deprivation in women to be even stronger with more recent data. First, older, as well as, more recent literature has shown that poor women or those living in disadvantage are more likely to develop negative health outcomes, while men less so.[23, 46, 58] Second, women are increasingly taking on multiple roles in society, such as, income-earner, child-bearer, and carer, which is adding to their burden (especially if they are living in deprivation).[24] Third, research has also shown that anxiety rates have been increasing in women in recent times.[59] For these reasons, we expect the association between area deprivation and generalized anxiety disorder to be even stronger in women at the present time.”

2. Comment:

Why categorise IMD to quartiles not quintiles as is standard?

Response:

As per comments from reviewers, we have now removed the analysis using the IMD and focused solely on the Townsend index to improve clarity and consistency in the presentation of our results.

3. Comment:

Original population 77,630; 30445 consented: 20,921 completed follow up HLEQ; 18494 eligible for analysis- 18494/30445+60% follow up so potential for selection bias. Would need a demographic table to see if those responding different from original cohort. No mention of this selection bias in the limitations section.

Response:

Thank you very much for this comment.

We have now included a section in the Limitations on this and, as requested, provided a demographics table comparing those who consented and refused to participate in EPIC-Norfolk using the information available to us (now included in Appendix I). The association we found within the cohort is unlikely to be explained by selection bias. It is unlikely that the association in non-responders would be in the opposite direction to that of responders. We now mention this in the Discussion.

4. Comment:

The literature on mechanisms for the association in women and deprivation described here is extensive in relation to Common Mental disorders and should be quoted more rather than the focus on environmental issues (ie educational attainment).

Response:

Thank you very much for this comment. As requested, we have now included a longer, comprehensive section in the Discussion on this.

We reviewed the literature again, and we elaborated on the mechanisms or the reasons why women living in deprivation might be more susceptible to common mental disorders.

This is what we added in:

“If women perceive their neighbourhood to be unsafe, they tend to restrict leisure activities, such as walking, and this can have further negative effects for their mental health.[48, 53] Perceiving neighbourhoods as unsafe can also erode social cohesion and can make women more hesitant to create social ties with others.[21] This can increase their risk of depression and related mental disorders, because women derive health benefits from being embedded in social networks.[21] Living in deprivation can also make individuals feel excluded from society and ashamed[54], and these feelings of exclusion are particularly harmful for women’s mental health[21, 54].”

We also added in a summary of recent studies examining the association between area deprivation and mental disorders from a gendered perspective, as indicated. We referenced recent, large studies with good research designs.

This is what we further included:

“Although other studies have shown that the places where people live have a substantial impact on health[15, 16], studies on the links between area deprivation and mental disorders from a gendered perspective are limited. A recent, large, population-based study[18] of over 21,000 people living in Ireland showed that area deprivation was associated with a significantly increased risk for common mental disorders in women, but not in men, after controlling for demographic and socioeconomic factors, and disability. In line with this, a study[21] of over 2700 adults living in Canada showed that greater neighbourhood disadvantage also was associated with increased risk of depressive

symptoms in women, but not in men. Research conducted in the US had similar findings.[54] This indicates that characteristics of the living context seem to influence women's health in particular."

Here are further explanations that we added to another part of the Discussion:

"Among disadvantaged women, it is not low income in itself that leads to poorer health, but rather the inability to derive the necessary resources from the environment to manage daily life; this can translate into stress and anxiety.[65] Women are becoming financially independent as they enter the labour force, which means that economic hardship now impacts them, as well. Women perceive economic hardship as a barrier to managing daily life and making ends meet, which can increase their anxiety. In contrast, men link job loss to a decline in social status."[45, 57, 65]

We had to be concise in our explanations and touched on key factors mentioned in the literature, because of word count considerations.

5. Comment:

There should be a lot more discussion of the negative finding in males- this is possibly out of keeping with what you would expect.

Response:

We have now elaborated in certain sections as to why we might not have seen this association in males – the explanations were based on the literature and were kept concise, because of word count restrictions.

This is what we added to our previous explanations for males:

"Women who are highly distressed tend to develop internalising disorders, while men are more prone to substance abuse and antisocial personality disorder.[56] The National Epidemiologic Survey on Alcohol and Related Conditions (NESARC) study[57] showed that total number of stressors experienced in life had a significantly stronger association with heavy drinking in men than in women. Therefore, men living in deprivation might be more likely to develop negative outcomes, such as, heavy drinking, rather than anxiety."

AND in another section, we further added:

In contrast (to women), "men link job loss to a decline in social status.[45, 57, 65] When men experience job-related stresses, they tend to externalize the effects of such stress and develop substance abuse.[57, 59]"

These were key factors raised in the literature.

6. Comment:

Why this method and not multi level analysis?

Response:

We used generalized estimating equations, because we were interested in determining the population average effect of area deprivation on the risk of developing generalized anxiety disorder. We believed that the intra-cluster correlation was very small – this was corroborated when we conducted logistic regression (results from the model based on logistic regression were identical to the model using generalized estimating equations).

VERSION 2 – REVIEW

REVIEWER	Martina Behanova Faculty of Medical Sciences Antonius Deusinglaan 1 9713 AV Groningen The Netherlands
REVIEW RETURNED	02-Feb-2017

GENERAL COMMENTS	Thank you for revised version. I do not have any other comments.
--